# Model-guided combinatorial optimization of complex synthetic gene networks

Joerg Schreiber, Meret Arter, Nicolas Lapique, Benjamin Haefliger & Yaakov Benenson[*] 

## Abstract

Constructing gene circuits that satisfy quantitative performance criteria has been a long-standing challenge in synthetic biology. Here, we show a strategy for optimizing a complex three-gene circuit, a novel proportional miRNA biosensor, using predictive modeling to initiate a search in the phase space of sensor genetic composition. We generate a library of sensor circuits using diverse genetic building blocks in order to access favorable parameter combinations and uncover specific genetic compositions with greatly improved dynamic range. The combination of high-throughput screening data and the data obtained from detailed mechanistic interrogation of a small number of sensors was used to validate the model. The validated model facilitated further experimentation, including biosensor reprogramming and biosensor integration into larger networks, enabling in principle arbitrary logic with miRNA inputs using normal form circuits. The study reveals how model-guided generation of genetic diversity followed by screening and model validation can be successfully applied to optimize performance of complex gene networks without extensive prior knowledge.

**Keywords** library screening; miRNA sensor; modeling; synthetic gene circuit
**Subject Categories** Network Biology; Synthetic Biology & Biotechnology
**Mol Syst Biol. (2016) 12: 899**

## Introduction

Optimizing quantitative characteristics of complex artificial gene pathways, networks, and circuits has been a long-standing problem in genetic engineering and synthetic biology. The bulk of the experimental optimization effort has focused on biomanufacturing pathways. Strategies included rational forward design of genetic components as well as component reshuffling followed by screening (Temme *et al*, 2012; Zhang *et al*, 2012; Jeschek *et al*, 2016). In this case, the optimization task is facilitated by the fact that in metabolic pathway optimization, the statement "the more the better" usually applies, achieved by concurrent optimization of pathways yield (ratio of product to substrate), specific productivity (product/cell per unit time) and volumetric productivity (product per unit volume per unit time). Sometimes these parameters can be anticorrelated (Villaverde *et al*, 2016), in which case the yield would typically take preference over volumetric and specific productivity (Sven Panke, personal communication). In synthetic multi-gene networks that implement regulatory or biosensing tasks, the optimization is exacerbated by the fact that no single readout can adequately characterize a system. Even in a single input/single output biosensor, performance is characterized by at least two parameters, the sensor response in the absence of an input (Off state) and the response with saturating input (On state), with the ratio between the two known as "dynamic range". Thus, a good circuit candidate needs to fulfill multiple conditions simultaneously. Directed evolution was used to improve circuit performance (Haseltine & Arnold, 2007; Schaerli & Isalan, 2013; Benes *et al*, 2015), but so far experimental results are limited to simple systems (Yokobayashi *et al*, 2002; Ellefson *et al*, 2014) or subcircuits (Lou *et al*, 2010).

Computational tools have played increasingly important roles in rational design of optimally performing circuits (Marchisio & Stelling, 2009). One can distinguish two complementary modeling approaches. An approach one might call "parameter-centered" uses mechanistic models to interrogate the parameter space or sensitivity to parameter changes. This allows identifying, respectively, parameter regimes that ensure (optimal) performance, and changes in parameters that may improve performance (Elowitz & Leibler, 2000; Gardner *et al*, 2000; Feng *et al*, 2004; Batt *et al*, 2007). This approach typically does not prescribe the genetic components that would implement the predictions. The second approach, which might be termed "component-centered", uses mechanistic models to predict behavior of complex networks built of known components whose basic features had been measured or predicted with relatively high precision (Ellis *et al*, 2009; Mutalik *et al*, 2013; Nielsen *et al*, 2016). When large-enough libraries of components with diverse and known behaviors are available, and the model correctly captures higher-order interactions that take place in a large network, the behavior of the large network can be (i) predicted with high precision for a particular set of components and (ii) tuned by choosing appropriate components from a component library. In reality, the parameter- and component-centered approaches are tightly interconnected, as the increasing availability of characterized components enables implementation of parametric model recommendations, while circuit construction and comparison

Department of Biosystems Science and Engineering, Swiss Federal Institute of Technology (ETH Zürich), Basel, Switzerland
*Corresponding author. Tel: +41 61 387 3338; E-mail: kobi.benenson@bsse.ethz.ch

of experimental data to a model allow model refinement to capture higher-order, long-range effects. An important prerequisite to the component-centered approach is prior knowledge of component properties and associated parameter values. Barring direct experimental characterization, predicting parameter values from *de novo* DNA or RNA sequence, while constantly improving (Zuker, 2003; Beisel *et al*, 2008; Salis *et al*, 2009; Choi *et al*, 2012; Rodrigo *et al*, 2012; Carey *et al*, 2013; Zhou *et al*, 2015), is still far from encompassing every aspect of molecular biology crucial for forward circuit design. Often the predictions are an outcome of high-throughput experiments followed by machine learning and are therefore specific to those experimental systems where data had been collected (Kudla *et al*, 2009; Egbert & Klavins, 2012; Alipanahi *et al*, 2015). Therefore, designing an optimally functioning circuit without a pre-existing library of experimentally characterized components is still a difficult task.

## Rationale

Here, we describe a novel approach that establishes an integrated computational-experimental framework for circuit optimization without extensive *a priori* knowledge and without a large pre-existing component library (Fig EV1). First, a parameter-centered computational analysis of a circuit is performed based on our best understanding of circuit's biochemical mechanism. The model predicts parameter regimes that optimize performance, as well as performance sensitivity to changes in individual parameters. Second, each circuit functional block is initialized with at least two or three functionally identical but structurally distinct genetic components, for example, two different transactivators, three different arrangements of miRNA binding sites, two different constitutive promoters, and so on. Where possible, the blocks are deliberately chosen to enact a desired change in a parameter value. Third, every possible combination of these components is tested; this is done to avoid the "guesswork" as much as possible and to account for possible errors, nonlinear effects, and higher-order interactions in a complex circuit that are not captured by the model. In addition, a dataset resulting from a combinatorial screening can be used either to validate or modify the model in the case of discrepancy between the two. The model is further validated by very detailed, low-throughput characterization of well-performing and poorly performing circuits. To summarize, at the end of an optimization campaign, several goals are reached simultaneously: The model receives experimental support (or modified to explain the data) such that it can be used to guide further experimentation; one or more well-functioning circuits are constructed; and the sets of initially tested building blocks can be used as reference points to construct additional components.

While a combinatorial screen can in principle be done without a model, such a screen will miss out on many important aspects: First, the initial library might not be optimally designed without the knowledge of how specific parameters affect performance; second, one might not be able to rationalize the results and explain why certain circuits perform better than others; and third, no rational conclusions will be drawn to serve subsequent design tasks. In other words, the model "bookends" the process: It serves as a formal system description and as a (partial) guide for library design; and at the end of the experimental campaign, it is validated and possibly modified to guide future design efforts.

Here, we explore this optimization strategy using a low-footprint proportional miRNA sensor as a test bed. The feasibility of such sensors was shown recently (Lapique & Benenson, 2014), but initial efforts to implement them practically resulted in poor performance. To address the problem comprehensively, we build on the extensive *in silico* study of a mechanistic model (Mohammadi *et al*, 2017) that provides specific recommendations with respect to optimal parameter regimes, followed by construction of a diverse genetic component library implementing some of the model recommendations, and exhaustive library screening using cell culture robots and imaging-based characterization. We uncover a number of high-performing sensors and undertake detailed mechanistic studies, confirming that those sensors perform well for the reasons predicted by the model. Lastly, we use the model to guide the construction of optimized sensor networks able to compute universal logic with miRNA inputs as exemplified by a difficult-to-implement "exclusive or" (XOR) circuit.

# Results

### Setting up the screen

Proportional miRNA sensors are key building blocks of miRNA sensing networks and miRNA cell classifiers (Xie *et al*, 2011; Lapique & Benenson, 2014; Li *et al*, 2015; Miki *et al*, 2015; Sayeg *et al*, 2015; Wroblewska *et al*, 2015). The cell classifiers can enable selective cell targeting in cancer and genetic disease, as well as selective detection and diagnostic tools. Published variants of these sensors, while showing exceptional dynamic range, also require large DNA payload. Previously, we found (Lapique & Benenson, 2014) that the protein repressor component of this sensor is in principle dispensable, potentially resulting in a more compact design (Fig 1A). In this sensor, an output gene driven by a constitutive promoter is repressed by an artificial miRNA molecule via complementary target sites in its untranslated region. The artificial miRNA is itself transcriptionally induced by a constitutively expressed transcriptional activator. The activator is targeted by the input miRNA via complementary target sites embedded in its untranslated sequence. As a result, in the absence of the input, the activator induces the synthetic miRNA, which in turn suppresses the output. In the presence of the miRNA input, the activator is knocked down, expression of the regulated synthetic miRNA is reduced, and output expression is elevated. Such compact topology might be used in low-capacity viral vectors and potentially deployed *in vivo*. Another change we considered was the use of constitutive activator rather than Dox-inducible rtTA (Xie *et al*, 2011), to enable fully autonomous sensor operation. However, initial tests with this topology using constitutive tTA activator in HeLa cells showed low dynamic range and poor recovery of the On state (Fig EV2) relative to the constitutive output. We explored ways to optimize sensor performance as a part of theoretical analysis, using a simplified mechanistic model of the compact sensor (Fig 1A and Materials and Methods) (Mohammadi *et al*, 2017). In the model, there are three parameters with the superscript MAX, corresponding to what we call the "pool" of the respective species.

                    

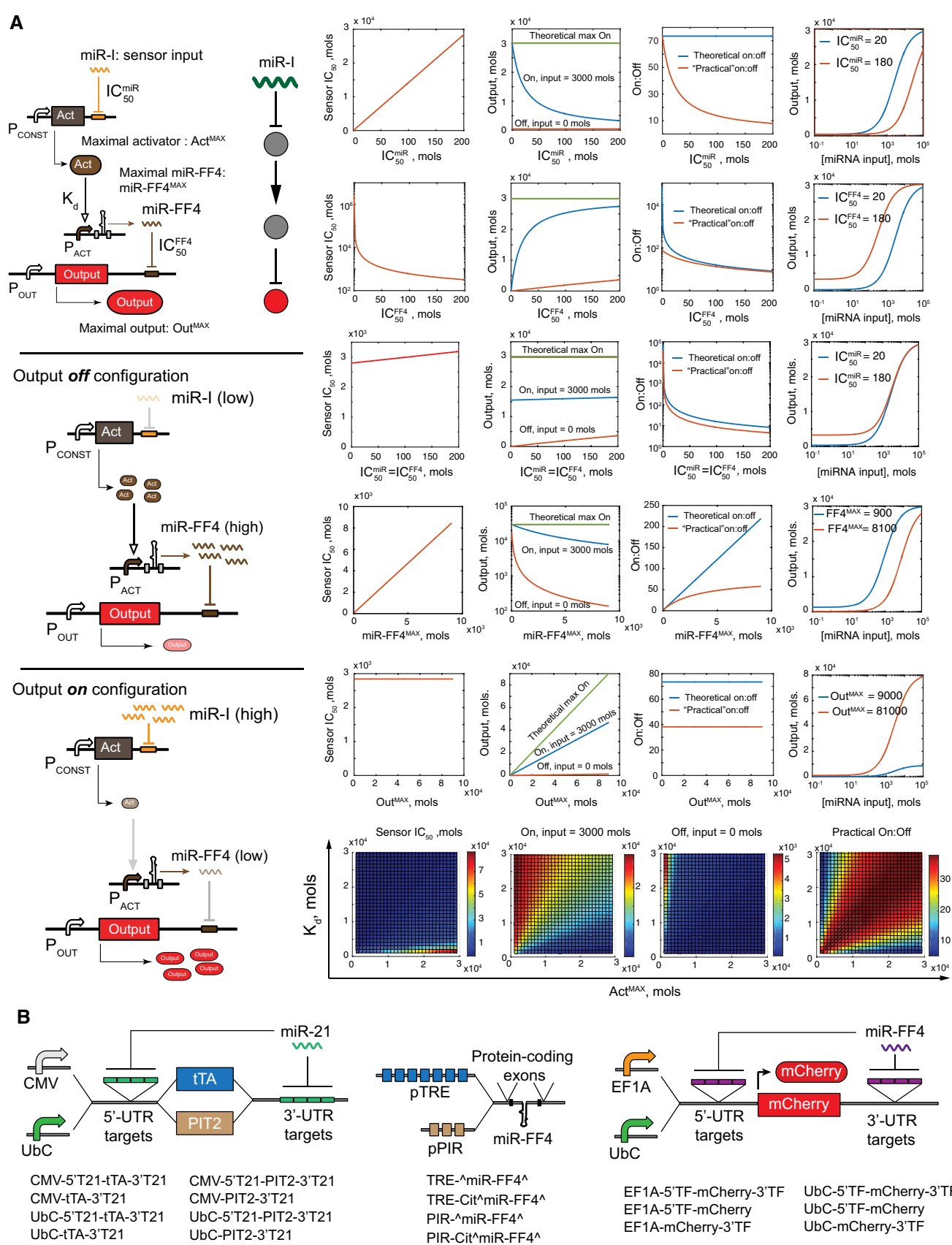

Figure 1.

**Figure 1.  Sensor schematics.**

A    Left upper panel: General architecture of a compact proportional miRNA sensor. A constitutively expressed activator Act regulates a synthetic miRNA miR-FF4 that in turn downregulates constitutively expressed output. Left middle panel: In the absence of endogenous miRNA input (miR-I), activator is highly expressed, miR-FF4 is highly expressed, and the output is repressed. The network diagram shows the topology implemented by this sensor. Left lower panel: In the presence of miR-I, the activator and miR-FF4 are low and the output is high. The parameters used in the model for simulation are indicated next to their respective species or interactions (see main text for explanation). Right: Model predictions regarding changes in On and Off sensor states as a function of parameter values. For each row (apart from the bottom one), the first three charts show, respectively, the changes in sensor sensitivity, On and Off states, and the dynamic range. The fourth chart shows sensor response function as a function of input concentration for two different parameter values, to exemplify the effect. In the bottom row, the heat maps show the effect of the activator pool size and the activator dissociation constant on the sensor sensitivity, On and Off states, and the dynamic range.

B    The structure of the combinatorial library utilized in the sensor screening campaign. Fixed components are embedded in the respective constructs, while interchangeable components are shown as branching units. On the right, the names of different constructs are shown, as used throughout the article. In the output construct names, Cherry is sometimes omitted for brevity because all the outputs use the same protein. CMV, cytomegalovirus promoter. UbC, ubiquitin C promoter. EF1A, elongation factor 1A promoter. tTA, tetracycline-controlled transcriptional activator. PIT2, pristinamycin-dependent transactivator. pTRE, tTA responsive DNA element. pPIR, PIT2 DNA binding motif. 5′ and 3′-UTR stand for 5′ or 3′ mRNA untranslated regions, respectively. T21: quadruple fully complementary miR-21 target; Cit: citrine exons; ^miR-FF4^ indicates an intronically embedded miR-FF4. TF, a triple fully complementary miR-FF4 target site.

For the activator and the output, the pool corresponds to the uninhibited expression levels of these species. For miR-FF4, the pool is its maximal asymptotic expression when the inducible upstream promoter is fully activated. There are also three different parameters that describe regulatory interactions (the lower they are, the stronger the interaction is for a given amount of regulator). They are $IC_{50}^{miR}$, $K_d$ and $IC_{50}^{FF4}$ corresponding, respectively, to the input miRNA concentration that elicits 50% knockdown of the activator, the dissociation constant of the activator from its promoter, and the level of miR-FF4 resulting in 50% knockdown of the output. Specific predictions and trends are summarized in Fig 1A. Note that we distinguish two "On" states. The theoretical On state is equal to the available output pool, because under asymptotically large amount of input the amount of the activator, as well as that of the synthetic miR-FF4 asymptotically approach zero. The "practical" On state is calculated using a finite, realistic amount of miRNA input, corresponding to a highly expressed cellular miRNA species (3,000 molecules/cell). For each parameter combination, we also calculated the overall sensor sensitivity we call Sensor $IC_{50}$, corresponding to the amount of miRNA input that elicits 50% of the theoretical maximal sensor response.

The simulations generate a number of specific predictions, in particular regarding expected trends in sensor On and Off states, and consequently, the respective dynamic range On:Off (Mohammadi *et al*, 2017). Increase of $IC_{50}^{miR}$ (that is, decrease in miRNA activity toward its target) results in decreased sensor sensitivity (higher Sensor $IC_{50}$ values), decreasing "practical" On state without change in the Off state, and corresponding decrease in dynamic range. Increase in $IC_{50}^{FF4}$ causes changes in exactly the opposite direction, resulting in increased sensor sensitivity, as well as increase in both On and Off states, yet overall decrease in dynamic range. Interestingly, when both miRNA regulation parameters are increased simultaneously, some trends cancel each other so the sensor sensitivity and the On state are roughly constant (Fig 1A, row 3). However, the Off state increases, resulting in a decreased dynamic range. Overall, it is predicted that decreasing both parameters improves dynamic range.

Increasing the miR-FF4 pool linearly reduces sensor sensitivity, while at the same time increasing the dynamic range. The practical dynamic range improves asymptotically. Here, a balance should be found between high dynamic range and high sensitivity, as these features are anticorrelated. The output pool does not affect sensor sensitivity; it affects both On and Off states linearly and therefore

does not affect the dynamic range. However, the output pool plays a key role when specific absolute output levels are required.

The size of the activator pool and an activator dissociation constant have a complex effect of sensor performance. Increasing the pool will result in decreasing On and Off states, but there exists an optimal pool size that maximizes dynamic range. Likewise, for a fixed pool, the On and Off states increase with higher dissociation constant (weaker binding), yet there is a particular $K_d$ value that maximized the dynamic range. In the phase space of activator pool size and $K_d$ values, there exists an optimal crest corresponding to a particular ratio of these two parameters.

Based on previous observations (Lee *et al*, 2009), we hypothesized that miRNA inhibition can be rationally improved by placing miRNA binding sites in both the 5′- and 3′-UTR of a targeted gene. We built our sensor library to detect miR-21, a known onco-miR (Selcuklu *et al*, 2009). In the context of activator components, we compared the constructs with 3′-UTR miR-21 binding sites (T21) with those carrying T21 sites in both 5′- and 3′-UTR (Fig 1B). Model recommendation regarding activator expression and activator binding constant is more difficult to implement rationally. One can modulate the binding constant by choosing different activator proteins (as the specific molecular nature of the activator does not affect circuit function). To modulate maximal expression rate, one can choose different constitutive promoters of different expression strength. It is plausible to assume that when many possible activator–promoter combinations are tested, at least one combination will fulfill the model recommendation. Thus, we used two very different components for the activator itself, tTA (Gossen & Bujard, 1992) and PIT2 (Fux & Fussenegger, 2003), and two different promoters, CMV and UbC, the former driving stronger expression than the latter. Together with two miRNA target configurations, this resulted in a total of eight activator constructs (Fig 1B, left). For the synthetic miRNA component, we used a proven miRNA sequence miR-FF4 (Leisner *et al*, 2010), furnished with two different inducible promoters, TRE or PIR, to respond to tTA or PIT2 activators, respectively; we compared a construct with miR-FF4 coding intron only, with a construct where the intron was embedded into a protein-coding mCitrine gene as the means to tune miR-FF4 expression. For the output, we compared two different constitutive promoters, EF1A and UbC; and three different miR-FF4 target (TF) arrangements, 5′-UTR only, 3′-UTR only and 5′- with 3′-UTR TF targets, to modulate miR-FF4 activity toward the output (Fig 1B, right).

    

## Screening results

The construct diversity resulted in a total of 96 sensor compositions. We used HeLa cells to measure their performance. Highly expressed endogenous miR-21 in these cells was used to estimate sensor On state, while co-transfection of LNA miR-21 inhibitor (LNA-21) was used to estimate the Off state. Triplicate measurements of On and Off states, combined with control measurements, resulted in about 1,000 individual transfections. One sample out of each triplicate was also measured with flow cytometry to confirm the imaging-derived data. The image-processing data in general correlate well with flow cytometry data (Fig EV3A) apart from some discrepancy in estimating very low Off states.

We examined the screening data for trends. First, we note that different outputs have different expression levels (Fig EV3B), and the level is slightly reduced when these constructs are combined with the miRNA-expressing cassettes possibly due to leakage of the inducible promoters in the absence or upstream activator. The highest possible sensor output is obtained when the activator construct is fully inhibited by a miRNA input; this level is emulated in a control experiment when the activator construct is entirely omitted from the transfection. Model predictions (apart from those testing output effect explicitly) assume that the output pool does not change; therefore, it is appropriate to test model predictions when different data points are normalized by the relevant output pool. Nevertheless, absolute levels could be important, and therefore, we present the data with and without normalization, as indicated. We note that the dynamic range is insensitive to output normalization.

One can observe the trends visually using heat maps (Figs 2A and EV2C) or by analyzing averaged effects between pairs of sensors that only differ in one component, or at most two when the activators are exchanged (Fig 2B). In the heat maps, one can directly observe the specific changes for particular sensor compositions while the average effects show the trend for all sensors in the set. In general, these two ways of comparison lead to similar conclusions. Thus, we observe that reducing activator expression with UbC promoter increases both the On and the Off states, as predicted by the model. The dynamic range increases slightly, which is not inconsistent. Second, improving miRNA knockdown of the activator using 5′- and 3′-sites is also fully consistent with the model: The On state increases while the Off state remains constant. The change in the nature of an activator is more complex to rationalize as two factors change at once: the binding constant and the activator pool (due to possibly different mRNA and protein stability of these species). The fact that the On state is not changed suggests that one of the activators has higher abundance and lower binding affinity than the other. This is consistent with published data and our own observations: Western blot analysis reveals that CMV- or UbC-driven PIT2 FLAG-tagged constructs are expressed around two orders of magnitude stronger than their respective tTA counterparts (Fig EV4). Published data also suggest that the binding of TetR to its operator is about 100 times stronger than that of PIT2 (Orth *et al*, 2000; Folcher *et al*, 2001). Adding an exon (indicated as Cit^miR-FF4^) arguably reduces the amount of active miRNA-FF4 due to slower transcription. Data show that the On state increases somewhat while the Off state increases dramatically, exactly as predicted. The dynamic range is therefore reduced, also

as expected. Using stronger output promoter results in comparable increase of absolute On and Off values, while this difference disappears post-normalization. There is only a weak effect on the dynamic range, as predicted. Different target arrangements in the output, either 5′- alone, or 5′- with 3′-combination, lead to more or less constant On state but substantially increased Off state. This would be consistent with the prediction if we assumed that these modifications in fact reduced the activity of miR-FF4 toward the output. Interestingly, a test in a different cell line (HCT-116) (Fig EV5) shows a reduction in On and Off states and an increase in dynamic range, suggesting that in this cell line, miR-FF4 targets in both 5′- and 3′-UTRs are beneficial compared to only 3′-UTR. It is also plausible that miR-FF4 is so efficient in HeLa cells that the 3′-UTR target alone already elicits maximal effect.

The averaged trends are consistent with the model prediction, but the screen also allows us to choose individual best-performing sensors. Here, based on dynamic range only, the preferable activator construct is CMV-tTA with 5′- and 3′-targets, combined with TRE-miR-FF4 without protein-coding exons, and either EF1a or UbC promoter driving the output with either 3′- or 5′+3′-targets. Combining this with HCT-116 data above, the combination of 5′- with 3′-targets is preferable, while the promoter choice depends on absolute output levels required under particular circumstances.

### Sensor follow-up and validation

A few selected sensors were re-measured using manual triplicate transfection and flow cytometry (Fig 3). The trends were consistent with the model predictions. For example, the only variable component between sensors **2** and **3** is the output promoter, and both the On and the Off states decrease when a weaker UbC promoter is used. Between sensors **3** and **4**, the Citrine exons are removed, resulting in increased miR-FF4 levels and further decrease in both On and Off states. Comparing sensors **3** to **5**, where a strong CMV promoter is replaced with a weaker UbC promoter, we observe increase in both On and Off states, as expected. Comparing sensors **6** to **3**, where the activator tTA is replaced with PIT2, we observe slight decrease in On and Off states. As discussed above, due to two simultaneous changes that occur during this replacement, the exact trend is difficult to predict. However, comparing sensors **7** to **6**, we once again confirm the increase in both On and Off states when a weaker promoter is used to drive an activator.

### Mechanistic verification

To examine whether performance improvements indeed happen for the reasons identified by the model, we performed detailed characterization of sensors **1** and **3** from Fig 3, which we denote as 3′-sensor and 5′+3′-sensor. We titrated the amount of input and measured all the relevant species on RNA and protein levels. tTA and mCherry mRNA, and miR-FF4, were measured by qPCR; tTA protein was quantified by Western blots; and Citrine and mCherry levels were measured by flow cytometry. MiR-21 levels were modulated with varying amount of LNA-21; the resulting miR-21 activity was measured using a bidirectional reporter (Fig 4A). In addition, one data point was obtained by exogenously transfecting miR-21 mimic into HeLa cells. We measured the dependency of different species in the sensor cascade on the miR-21 activity. As we progress

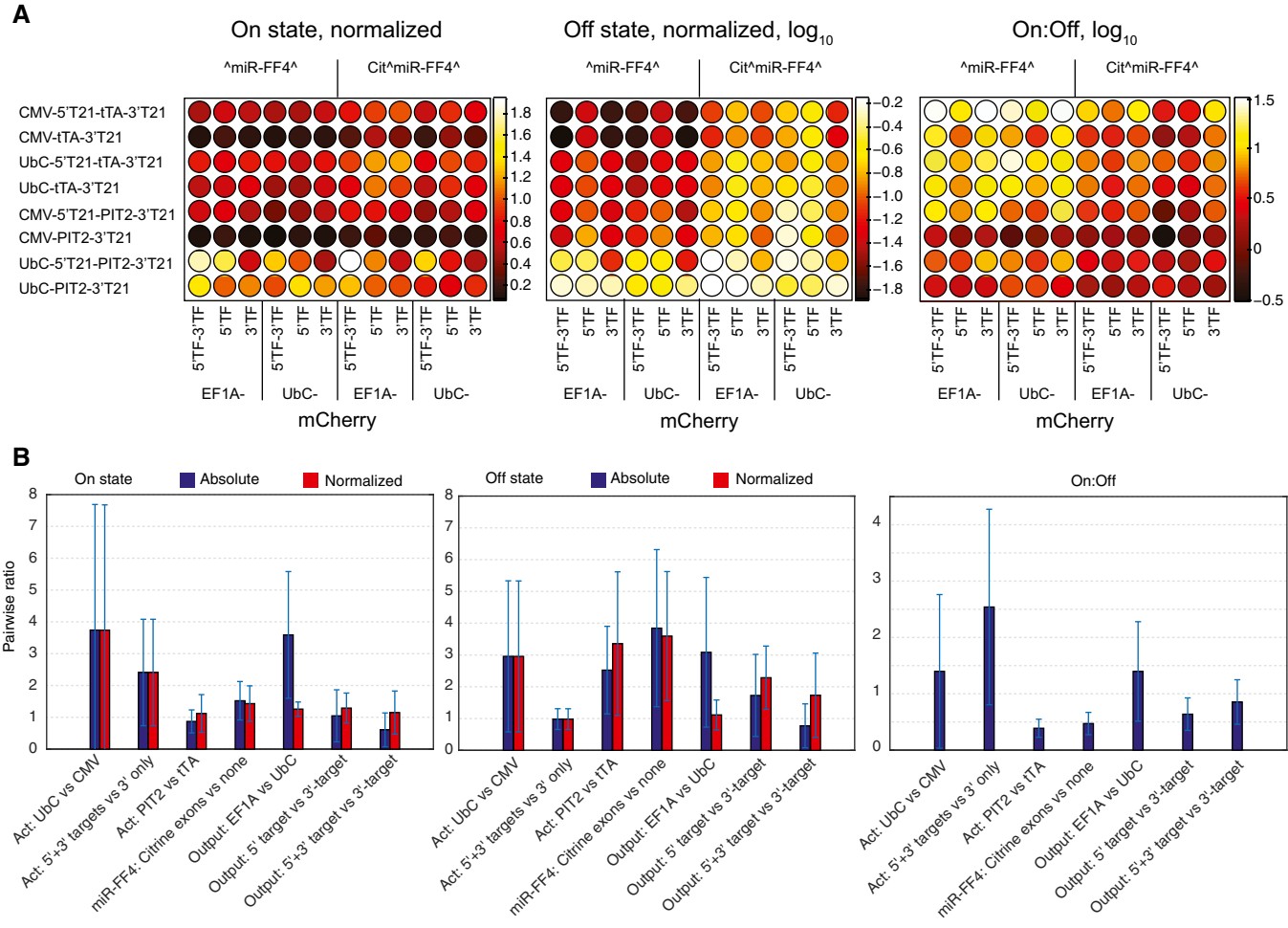

**Figure 2. Screening results.**

A  Heat maps depicting the experimental sensor performance data in the form of a 96-well plate. Each well represents one sensor composition. The maps show the On, log-transformed Off state, and log-transformed dynamic range (to emphasize differences at the low values). Sensors in each row use the same activator construct (as indicated); sensors in the first six columns employ the exon-less miR-FF4 construct and mCitrine-embedded miR-FF4 in the columns 7–12. Sensors in each column use the same output construct as indicated below. FF4 constructs are driven by TRE promoter when tTA is used and by PIR promoter when PIT2 is used. All values were measured in triplicates and normalized to the theoretical maximal output level obtained in the presence of miR-FF4 expressing cassette and the output, without the activator.

B  Bar charts showing trends in sensor performance, namely, changes in On states, Off states, and the dynamic range. The trends are calculated by pairwise comparison of sensors that only differ in a single building block, as indicated on the *x*-axis. In these labels, "X versus Y" means that the values measured with X are divided by the values measured with Y. Act, activator. For activator replacement, both the activator and its regulated promoter must be changed simultaneously. The trends for On and Off states are shown in absolute units (blue), or following pre-normalization (red) to the respective size of an output pool for each individual sensor. Screening experiments were performed in HeLa cells. The error bars represent standard deviation of the parwise ratios, 48 values in all cases apart from output target comparison where 32 values are used.

Source data are available online for this figure.

down the cascade, we make the following observations. First, the mRNA of tTA is almost not affected by miR-21 (Fig 4B); measurement noise is large and there might be a certain downward trend with the 5′+3′-sensor, but it is inconclusive. However, the bulk of the regulation occurs at the protein level (Fig 4C), despite the fact that we use fully complementary miR-21 targets. The response sensitivity, which we define as the amount of input eliciting half the overall effect ($IC_{50}$) for the 5′+3′-sensor, stands at about 5% of miR-21 activity, while it is around 23% for the 3′-sensor, consistent with expectation that 5′+3′ targets are more sensitive to miR-21. MiR-21 → miR-FF4 mapping exhibits $IC_{50}$ of about 20% for 5′+3′-sensor and

45–50% for 3′-sensor (Fig 4D). Comparable difference between $IC_{50}$ values is observed when we examine miR-21 → mCitrine dependency (Fig 4E). This difference is carried over from miR-21 → tTA mapping, because the transfer curve between tTA and miR-FF4 is similar for both sensors with $IC_{50}$ at 330 and 260 tTA protein units for 3′- and 3′+5′-sensor, respectively, when tTA → miR-FF4 curves are considered (Fig 4F), and 350 tTA units for both sensors when using tTA → Citrine curves (Fig 4G), as expected. At the output level, mCherry mRNA is not affected by miR-21 (Fig 4H), similar to the observation with tTA. miR-FF4 → mCherry and mCitrine → Cherry (Fig 4I and J) dependencies show miR-FF4 $IC_{50}$ of 0.051 and

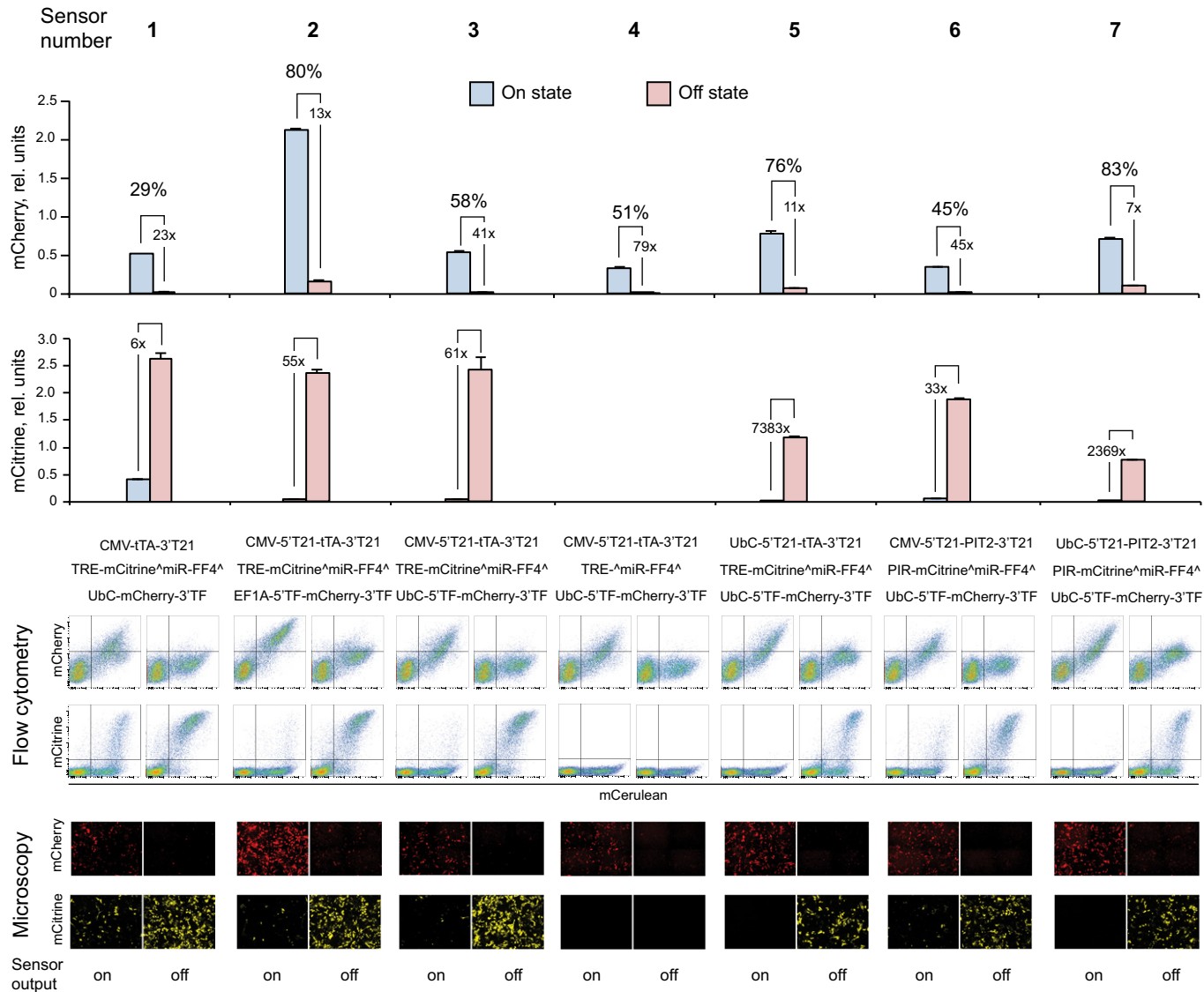

**Figure 3. Experimental validation of selected sensors.**

Sensors are numbered at the top. The bar charts show mCherry and mCitrine values of On and Off states, together with the fold-change and the percentage of recovery in the On state relative to the highest theoretical output. Sensor compositions are indicated below the charts, together with representative flow cytometry and microscopy images of triplicate measurements. Error bars represent standard deviations, and HeLa cells were also utilized in validation experiments. For construct notations, see the legend to Fig 1.

Source data are available online for this figure.

0.024 concentration units in 3′- and 3′+5′-sensor, respectively; and 0.31 and 0.18 units for Citrine. Thus, both measurements indicate higher sensitivity to knock-down when 5′- and 3′-targets are used, even though the difference is less pronounced compared to the knock-down of tTA by miR-21. For the cumulative input → output (miR-21 → mCherry) response, 3′-sensor exhibits $IC_{50}$ of 85% miR-21 activity and 5′+3′-sensor has $IC_{50}$ of about 55% miR-21, while at the same time sensor dynamic range increases from 21- to 46-fold (Fig 4K). To summarize, the incorporation of 5′- and 3′-miRNA targets in the activator and the output increases miRNA activity toward these targets. As we have observed, and consistent with the model, this results in modest decrease in $IC_{50}$ and large increase in the dynamic range.

**Compact sensor as the basis for universal miRNA logic**

Having identified a number of optimal sensor configurations, we asked whether the circuit is functional in additional cell lines and whether it can be reprogrammed to address different inputs. The miR-21 sensor **4** was tested in HCT-116 and HuH-7 cell lines that both express intermediate levels of miR-21. The data (Fig 5A and B) show that the sensor operates with good dynamic range in these cell lines. Next, the sensor was reprogrammed to sense miR-27 input by swapping the binding targets in the tTA gene. It was tested in HeLa cells in the presence of extra miR-27 (due to intermediate endogenous expression of this input) and upon miR-27 inhibition with LNA-27 (Fig 5C). The characteristics of the reprogrammed sensor

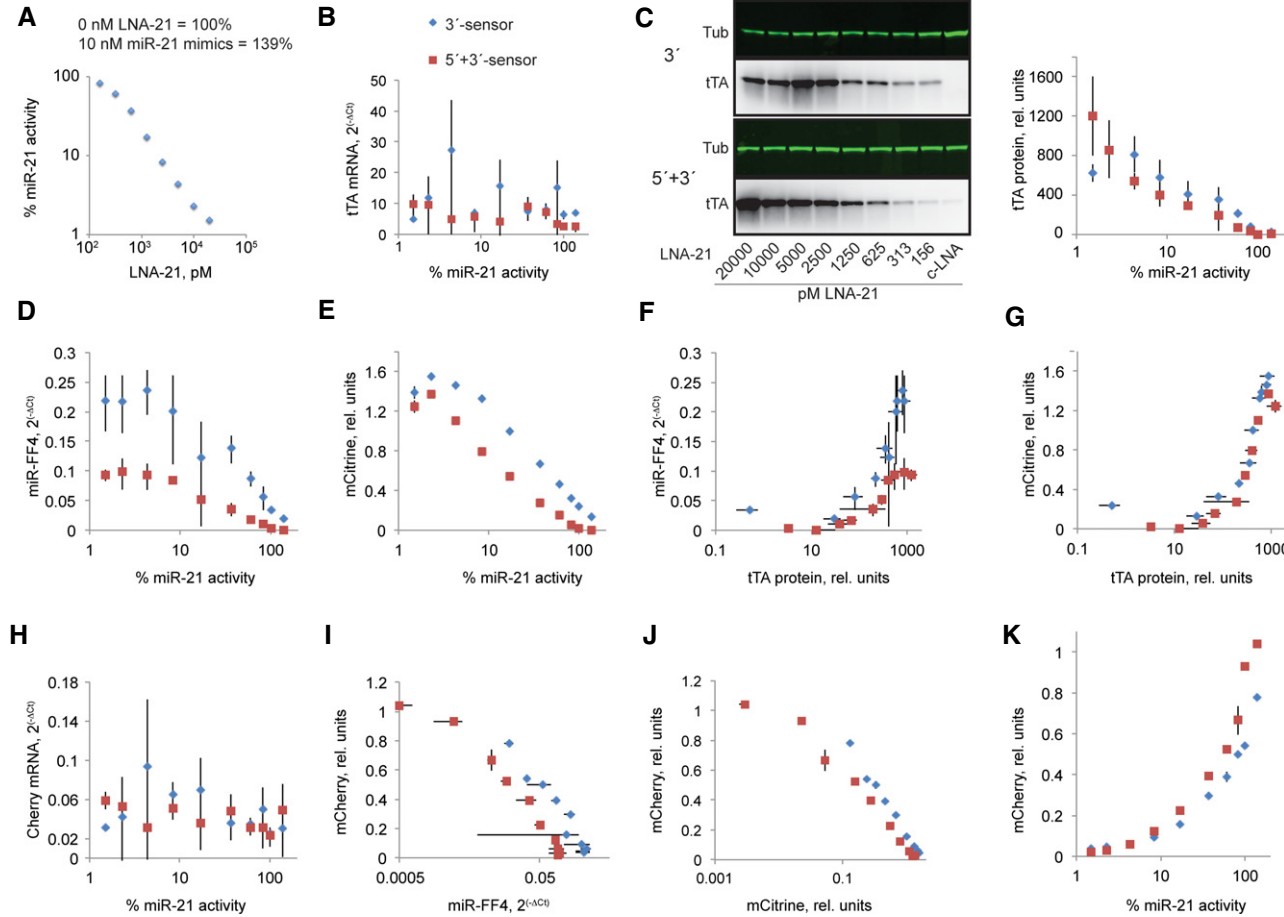

**Figure 4.    In-depth experimental mechanistic interrogation of two different sensors.**

A    The relationship between the amount of LNA-21 and the activity of miR-21 toward bidirectional activity sensor.

B    The relationship between miR-21 activity and mRNA levels of tTA as measured by qPCR. RNA expression is calculated using $\Delta C_t$ between tTA mRNA and that of a housekeeping gene.

C    miR-21→ tTA protein response curve. Left, representative Western blots.

D    miR-21 → miR-FF4 dependency. miRNA expression is calculated using $\Delta C_t$ between miR-FF4 and that of a housekeeping gene mRNA.

E    miR-21 → mCitrine dependency.

F    tTA → miR-FF4 dependency.

G    tTA → Citrine dependency.

H    miR-21 → mCherry mRNA dependency. RNA expression is calculated using $\Delta C_t$ between mCherry mRNA and that of a housekeeping gene.

I    miR-FF4 → mCherry protein dependency.

J    mCitrine → mCherry dependency.

K    miR-21 → mCherry protein dependency, embodying the input–output response of the sensors.

Data information: Data are the mean of triplicate measurements, and error bars are standard deviations. Experiments were done in HeLa cells. In panels (B–K), blue diamonds indicate Sensor 1 from Fig 3 while red squares indicate data for Sensor 3.

Source data are available online for this figure.

are very similar to those of the miR-21 sensor, suggesting that the architecture is flexible.

Sensors can operate in parallel if they utilize distinct sets of activators and internal miRNA molecules. Parallel operation of multiple sensors implements a core OR logic between their inputs. In addition, the outputs of each sensor can be targeted directly by miRNA inputs, amounting to AND logic between the positive input (targeting the activator) and negated inputs (targeting the output). The combination of AND, NOT, and OR logic operations enables in principle universal logic with miRNA inputs based on disjunctive normal

form. Despite earlier work on RNAi logic, universal computation has never been shown; instead, NOR (Rinaudo *et al*, 2007) and AND (Xie *et al*, 2011) gates were implemented. To illustrate the universal logic possibility, we implemented an XOR logic operation between two miRNA inputs that in the normal form expansion is equivalent to the relationship "[miR-A AND NOT(miR-B)] OR [NOT(miR-A) AND miR-B]". We already established that PIT2 can be used instead of tTA; in order to fully decouple the processing units, we replaced miR-FF4 with another synthetic miRNA, miR-FF6. We employed miR-21 and miR-27 as inputs, and constructed individual AND gates

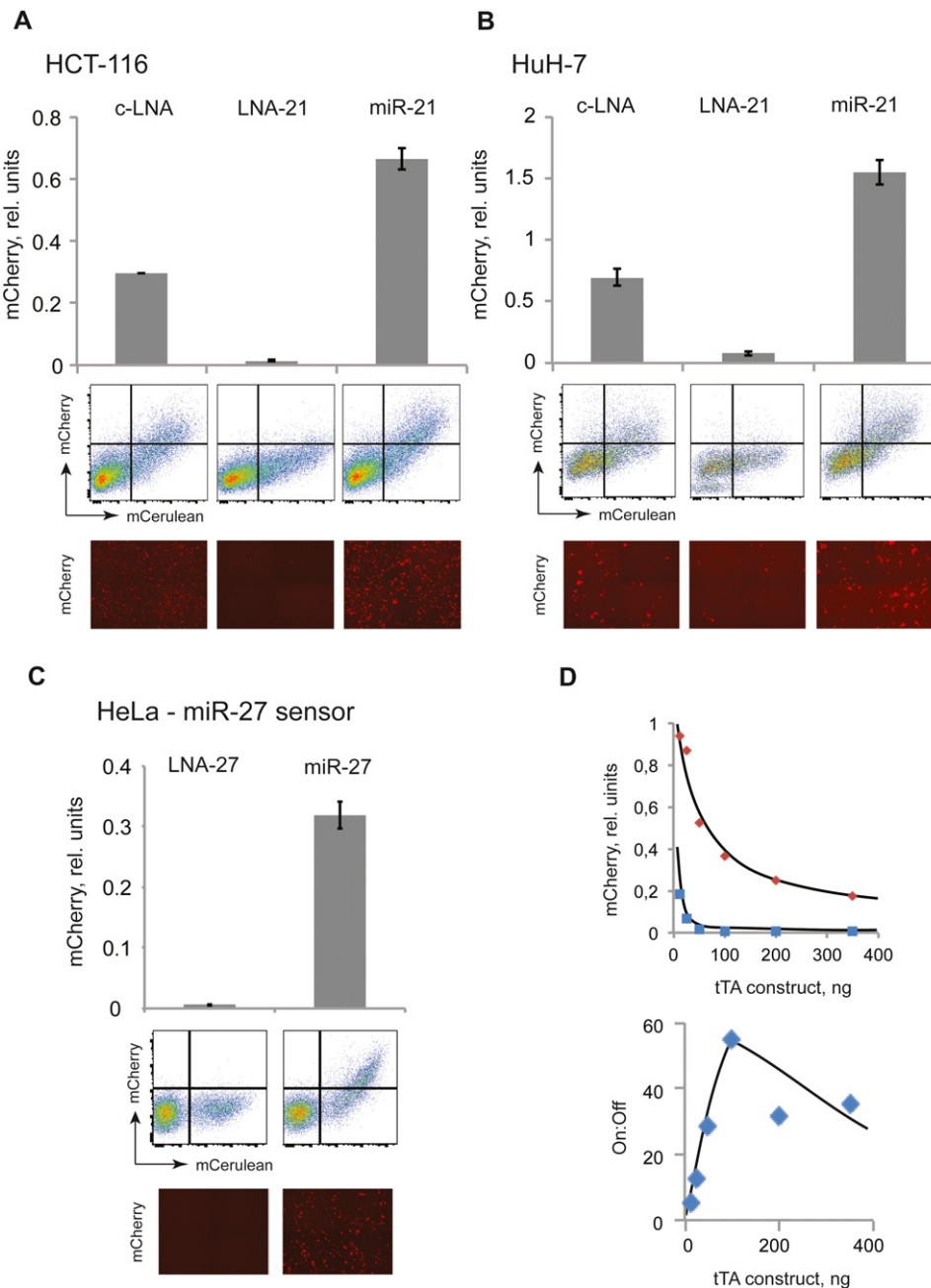

**Figure 5. Sensor portability, programmability, and fine-tuning.**

A, B    Sensor 4 (Fig 3) in HCT-116 (A) and HuH-7 (B) cell lines. We measured sensor output in response to endogenous miR-21 levels, upon miR-21 inhibition with LNA-21, and in the presence of extra miR-21 using mimic cotransfections.

C    Sensor reprogrammed to respond to miR-27 input in HeLa cells. Additional miR-27 is transfected to elucidate full sensor dynamic range.

D    Measured On and Off states and the dynamic range of Sensor 4 (Fig 3) with increasing activator dosage. The curves are drawn by hand to illustrate the trends

Data information: Bar charts of triplicate measurements and their standard deviations are shown together with representative flow cytometry plots and microscopy images.
Source data are available online for this figure.

"miR-21 AND NOT(miR-27)" using tTA and miR-FF4, and "NOT (miR-21) AND miR-27" using PIT2 and miR-FF6. However, our initial characterization of the PIT2/FF6 sensor showed low On values and low dynamic range. We used the model prediction regarding the effect of activator expression on the dynamic range, and reasoned that low On state could be rectified with the decrease in activator amount (Fig 1A), as supported by varying activator dosage with our "best case" sensor (Fig 5D). Thus, we reduced the amount of the PIT2 activator construct and improved the dynamic range substantially. These modifications allowed comparable performance of two "AND NOT" networks and the realization of the hard-to-implement XOR computation with miRNA sensors (Fig 6).

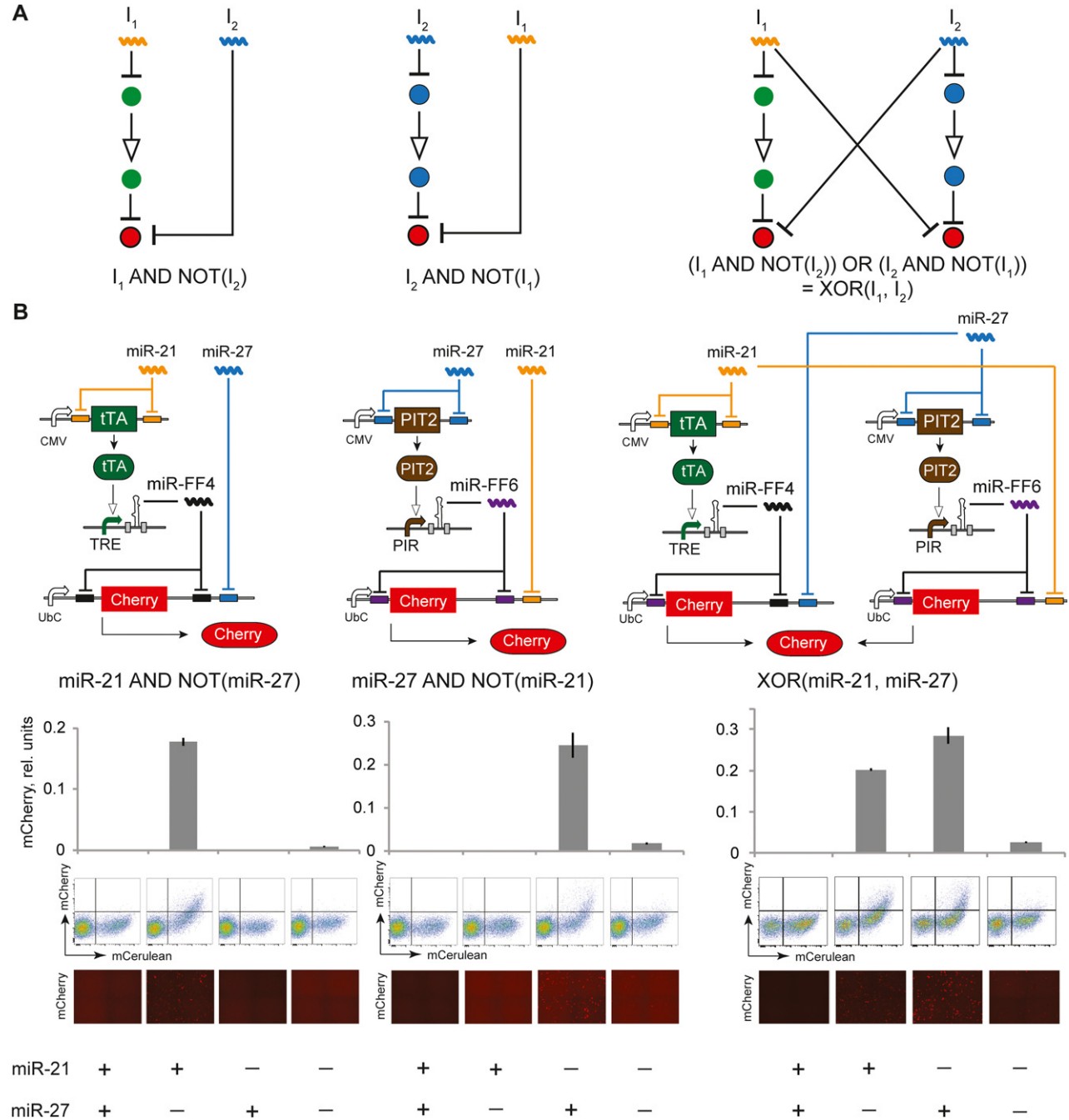

**Figure 6. Optimized compact sensor serves as the basis for universal logic with miRNA inputs.**

A Abstract network topology showing the "AND NOT" logic operations and the normal form expansion of the XOR logic function.

B Top to bottom: Molecular networks corresponding to the abstract networks; bar charts illustrating network response to all four input combinations; representative flow cytometry plots and microscopy images. mCerulean serves as transfection control. On state of a miRNA input is obtained with mimic co-transfection while Off state of a miRNA input is achieved using respective LNA inhibitor. The experiments were performed in triplicates in HeLa cells, and the error bars represent standard deviations.

Source data are available online for this figure.

## Discussion

In this study, we describe a novel strategy for the development and optimization of complex synthetic gene circuits. In general, optimal design of complex networks is hindered by two factors. First, the mechanistic model of a network might be incomplete and/or erroneous, resulting in wrong conclusions. If this is not the case, the model may correctly prescribe ways to improve system performance

via specific parameter adjustments, but not the concrete ways to implement these adjustments at the DNA level. The traditional approach to optimizing a genetic component would require repeated cycles of mutagenesis and selection, or prior experimental characterization of a large library of components to evenly cover the parameter range. Both approaches are very time-consuming. In our approach, we choose two to three basic component variants (i.e., promoters, activators, etc.) that are expected, based on the model and on our best knowledge, to access distant points in the parameter phase space. We then perform exhaustive combinatorial characterization of all possible combinations of these building blocks and use the resulting data to validate or modify the model and build solid foundation for further design efforts. Indeed, in the current study the generated diversity of 96 circuit variants already resulted in a number of well-performing circuits that far surpassed the performance of the original system. The analysis of trends (Fig 2B) confirms model predictions, although it is interesting to note that the agreement between the model and the data becomes apparent only when the data are considered in aggregate, judging from the large error bars in Fig 2B. In a detailed follow-up mechanistic study, we were able to confirm that the factors determining superior performance were consistent with the original model predictions. Lastly, the optimization study opened ways to reprogram the sensor to address additional inputs, setting the background for large-scale universal logic with miRNA inputs. The conclusions are also valid for any double-inversion topology that is homologous to our circuit, namely, a repressor input targeting an activator of a downstream repressor, in turn repressing the output (Fig 2A). The conclusion is that the inhibitory interactions must be as strong as possible with lowest possible $IC_{50}$, while the activating interactions should operate far from saturation (thus, its $IC_{50}$ should not be much lower than the highest anticipated activator level, obtained in the "Off" state).

This strategy can be generalized to additional multi-component systems where multiple operation parameters must be optimized simultaneously. Clearly, specific conclusions will tightly depend on circuit topology, and therefore they cannot be easily translated to other systems unless the latter already contain modules whose topology matches a previously investigated one. However, the workflow that begins with defining performance metrics, followed by computational mapping of the parameter space and identification of favorable regimes, further followed by a screen of combinatorially composed networks that access distant points in the parameter space, and concluding with model validation or modification, can be applied to any gene circuit. Even if the initial screen does not provide a desired solution, the dataset would usually be enough to guide a final round of focused optimization effort.

## Materials and Methods

### Modeling and simulation

The simulations were performed in MATLAB. The model describes the one-input proportional sensor (Fig 1A), and it is a special case of the general model (Mohammadi *et al*, 2017). Briefly, we assume non-cooperative Hill-like relationships between upstream and downstream components, as supported by published experimental data (Mohammadi *et al*, 2017). The concentration of the activator as the function of the miRNA input is described as:

$$[\text{Act}]([I]) = \text{Act}^{\text{MAX}} \frac{IC_{50}^{\text{miR}}}{IC_{50}^{\text{miR}} + [I]} = \text{Act}^{\text{MAX}} \left( 1 - \frac{[I]}{IC_{50}^{\text{miR}} + [I]} \right) \quad (1)$$

where [X] represent the concentration of molecular species X. [Act] is a steady-state concentration of the activator, either tTA or PIT2; $\text{Act}^{\text{MAX}}$ is the maximal steady-state activator concentration, without any RNAi knockdown. [I] stands for input concentration, here the miRNA sensor input. $IC_{50}^{\text{miR}}$ stands for miRNA concentration that elicits half the knock-down.

The equation governing miR-FF4 induction is:

$$[\text{miR - FF4}]([\text{Act}]) = \text{miR - FF4}^{\text{MAX}} \frac{[\text{Act}]}{K_{\text{d}} + [\text{Act}]} \quad (2)$$

where [miR-FF4] represents steady-state concentration of miR-FF4, [Act] is the activator level (computed with equation 1), $K_{\text{d}}$ is the apparent dissociation constant of the activator from its promoter. $\text{miR-FF4}^{\text{MAX}}$ is the maximal miR-FF4 expression from an inducible promoter under activator saturation.

Lastly, the output level is determined by the strength of miR-FF4 repression using a dependency similar to equation 1, namely,

$$[\text{Out}]([\text{miR - FF4}]) = \text{Out}^{\text{MAX}} \frac{IC_{50}^{\text{FF4}}}{IC_{50}^{\text{FF4}} + [\text{miR - FF4}]}$$
$$= \text{Out}^{\text{MAX}} \left( 1 - \frac{[\text{miR - FF4}]}{IC_{50}^{\text{FF4}} + [\text{miR - FF4}]} \right) \quad (3)$$

where [Out] is steady-state output concentration and $\text{Out}^{\text{MAX}}$ is the maximal output concentration in the absence of miR-FF4 knock-down. $IC_{50}^{\text{FF4}}$ denotes the activity of miR-FF4 toward the output.

For numerical simulations, we used the following basic parameter set:

$IC_{50}^{\text{miR}}$: 20 molecules/cell, $IC_{50}^{\text{FF4}}$: 20 molecules/cell, $K_{\text{d}}$: 10,251 molecules/cell, $\text{Act}^{\text{MAX}}$: 9,755 molecules/cell, $\text{miR-FF4}^{\text{MAX}}$: 3,000 molecules/cell, and $\text{Out}^{\text{MAX}}$: 30,000 molecules/cell.

These parameters were found to be optimal as well as plausible physiologically and correspond to parameters C1OR, C1NOT, C2, $T_{\text{max}}$, $FF4_{\text{max}}$, and $\text{Out}_{\text{max}}$ (Mohammadi *et al*, 2017).

When required, the basic parameters were varied as indicated in relevant figure legends or shown on graph axes.

Note: for a mammalian cell, 1 molecule/cell corresponds to ~ 1 pM (1,000 molecules/cell = 1 nM).

The simulations in Fig 1 were performed as follows: for each parametric scan, the varied parameters were tested in the range indicated in the plots. The fixed parameters were set at the values above. In order to calculate the $IC_{50}$ of the complete sensor, sensor input/output curves were generated in the range $10^{-1}$–$10^5$ input molecules and fitted to a Hill function. The resulting curves were fitted to the Hill function with leakage, namely, $y = b_3 + b_1 x / (b_2 + x)$. The fitted value of $b_2$ parameter was interpreted as sensor's $IC_{50}$.

In order to simulate an Off value, input value of 0 was substituted in the equations 1–3 above. For the On state, we considered two cases. The highest possible On state is equivalent to Out$^{MAX}$, that is, 30,000 molecules; however, the amount of input required to achieve this level could be extremely high. Therefore, we also evaluate something called "practical" On output that corresponds to 3,000 input molecules/cell. This value represents a reasonably highly expressed endogenous miRNA. Accordingly, the On:Off ratios are calculated once using the theoretical On state and once using the "practical" On state; the latter is likely to be characteristic of the experimental observations.

### Cloning

Plasmids were constructed using standard cloning techniques or synthesized by Genewiz. Restriction enzymes were purchased from New England Biolabs (NEB). Phusion High-Fidelity DNA Polymerase (NEB) was employed for fragment amplification. Primers were synthesized by Sigma-Aldrich (Table EV1). For agarose gel-mediated DNA purification, the GenElute Gel Extraction kit was used (Sigma-Aldrich). Ligations were performed using T4 DNA Ligase (NEB). Ligation products were transformed into chemically competent *E. coli* DH5α that were plated on LB Agar with appropriate antibiotics selection (ampicillin 100 µg/ml, kanamycin 50 µg/ml). Sequence integrity of the plasmids was confirmed by sequencing. In some cases, construct generation was based on previously published plasmids (Weber *et al*, 2002; Leisner *et al*, 2010; Xie *et al*, 2011; Prochazka *et al*, 2014; Angelici *et al*, 2016), see Table EV2 for additional information.

### miRNA-mimics and inhibitors

The following miR-mimics purchased from GE were employed: miR-27b (C-300589-05-0005), miR-21b (5081393), ctrl. (W9931K). As inhibitors, the Exiqon products with the catalogue numbers 4103307 (mir-27b), 4102261 (miR-21b), and 100006 (ctrl) were used.

### Cell culture

HeLa cells were purchased from ATCC (Cat # CCL-2) and cultured at 37°C, 5% CO$_2$ in DMEM, high glucose (Life Technologies, Cat # 41966), supplemented with 10% FBS (Life Technologies, Cat # 10270106), and 1% Pen/Strep solution (Sigma-Aldrich, Cat # P4333). Same medium and conditions were employed for HCT-116 cells (Clontech, Cat #630931) whereas HuH-7 cells, received from the Health Science Research Resources bank of the Japan Health Sciences Foundation (Cat #JCRB0403), were grown in GlutaMAX (Life Technologies, Cat # 21885-025), supplemented with 10% FBS (Life Technologies, Cat # 10270106) and 1% Pen/Strep solution (Sigma-Aldrich, Cat #P4333).

### Cell transfections

Cells were seeded 16 h before transfection at varying density in order to achieve an 80–90% confluence at the time of transfection in 24-well plates: HeLa 90,000 cells, HCT-116 200,000 cells, and HuH-7 50,000 cells per 24-well plate. OptiMEM-diluted DNA samples and Lipofectamine 2000 (Life Technologies) were combined and pipetted dropwise onto the cells after a 10-min incubation. miRCURY LNA power inhibitors or Dharmacon miR-RNA mimics were added before transfection as indicated. Based on the protocols for 24-well transfections, all DNA amounts and volumes were scaled by a factor of 0.2 for transfections in 96-well format for robotic transfections. A HAMILTON STARplus liquid handling workstation dedicated to the automated cultivation and transfection of adherent mammalian cell lines was used to set up the master plates and perform the transfection of HeLa cells. A customized script using the Venus application from Hamilton was developed to control liquid handling, pipetting, and mixing of the DNA solutions during the screening experiment.

### RNA preparation and qPCR

Cells were harvested using TRIzol according to the instructions of the manufacturer. Before RNA precipitation, glycogen was added as recommended in the protocol. RNA amounts were quantified using a Nanodrop, and equal amounts were reverse transcribed following a vigorous DNase digest (Ambion, Cat # AM1906). miR-FF4 was reverse transcribed using the Exiqon universal cDNA synthesis kit and amplified with customized primers by the Exilent SYBR Green master mix. For reverse transcription and amplification of protein-coding genes, gene-specific primers were designed. A list of primers can be found in the Table EV1. cDNA was generated by the Revert Aid Premium First Strand cDNA synthesis kit (Thermo) and amplified using the Light Cycler 480 Green I Master Mix (Roche).

### Fluorescence microscopy and flow cytometry analysis

Microscopy: 48 h after transfection, the cells were visualized for fluorescence using a Nikon Eclipse Ti Microscope provided with a Hamamatsu ORCA-R2 camera and controlled by the Nikon NIS-Elements software. For the screening experiment, a similar analysis strategy was pursued as in a recent publication (Haefliger *et al*, 2016). Flow cytometry: 48 h after transfection cells were harvested by incubation with 0.2 µl phenol red-free trypsin (0.5% trypsin-EDTA (Gibco, Life Technologies, Cat # 15400-054) at 37°C for 3 min. The prepared samples were analyzed using a BD LSR Fortessa II Cell Analyzer with a combination of excitation and emission that minimizes the crosstalk between different fluorescent reporters. mCherry was measured with a 561-nm excitation laser coupled with a 600-nm longpass filter and 610/20-nm emission filter, mCitrine with 488-nm laser, 505-nm longpass filter and 542/27-nm emission filter and mCerulean using a 445-nm excitation laser and 473/10-nm emission filter. FlowJo software was used for data analysis. The provided flow cytometry plot data are created through the FlowJo Layout editor, and the shown plots represent one sample out of a biological triplicate. In order to quantify the flow cytometry measurements, scores of the different channels were calculated by multiplying the frequency by the mean. To normalize for transfection efficiency, the score of the investigated fluorophore/reporter (mCherry or mCitrine) was normalized by independently expressed transfection control (mCerulean). The operations can be summarized in the equation:

fluorophore intensity in relative units (r.u.)

= [mean of fluorophore-positive cells

× frequency of fluorophore-positive cells]

/[mean of transfection ctrl. positive cells

× frequency of transfection ctrl. positive cells].

### Western blotting

Adherent cells were harvested after several consecutive washes by scraping in PBS supplemented with a protease inhibitor cocktail. Cells were lysed by adding 1× Laemmli sample buffer containing 62.5 mM Tris–HCl, pH 6.8, 2% SDS, 25% glycerol, 0.01% bromophenol blue, and 2-mercaptoethanol. Protein expression was analyzed by standard procedures for Western blotting using 12% Bio-Rad Criterion pre-casted gels and a Trans-Blot Mini device for protein transfer onto PVDF membranes. α-Tubulin and FLAG-tagged tTA or PIT2 expression was visualized with a fluorescent secondary antibody on a LI-COR Odyssey Clx whereas conventional ECL reagent providing a higher sensitivity was used to quantify tTA-FLAG expression with an Image Quant LAS 4000mini device.

### Antibodies

Detection of FLAG-tagged constructs was achieved using monoclonal ANTI-FLAG M2 as primary antibody (Sigma, Cat # F1804) and an HRP-linked secondary antibody (GE Healthcare, Cat # NA931). The signals were visualized with SuperSignalWest Femto MaxSubstrate (Life Technologies, Cat # 34095). For tubulin Sigma's T6199 monoclonal antibody was used and a fluorescently labeled secondary antibody visualized the signals (Li-Cor, Cat # 92632210). Quantification was done using ImageStudioLite software (Li-Cor Biosciences).

### Data analysis

In order to determine $IC_{50}$ values in Fig 4, the highest output value was set to 100% in knock-down curves (miR21→tTA, miR-FF4/Citrine→Cherry). The input value corresponding to 50% of the output was calculated via inverse mapping using linear interpolation of the data in the vicinity of this output value, using three data points in total with at least one point above 50% output and one below. In induction curves, that is, tTA→miR-FF4 and tTA→Citrine, output saturation is achieved; the highest output values in the data series were set to 100% and half-output values were calculated accordingly. In the curves with mCherry output, the 100% output value was measured in the control experiment where the activator was omitted; thus, there were no assumptions made regarding saturation in the response curves.

**Expanded View** for this article is available online.

### Acknowledgements

This study is funded by Swiss National Science Foundation, NCCR "Molecular Systems Engineering" and ETH Zurich. We thank Urs Senn from the Liquid Automation Facility for the development of automated transfection protocols. We thank T. Horn, T. Lopes, and V. Jäggin for their help with flow cytometry; and N. Beerenwinkel, P. Mohammadi, and the members of Benenson laboratory for discussions.

### Author contributions

JS, NL, and YB conceived the study. JS performed the great majority of experiments, analyzed data, and wrote the manuscript. MA and NL performed experiments at the early stage of the project and proofread the paper. BH analyzed data and proofread the paper. YB created the mechanistic model, analyzed data, wrote the paper, and supervised the project.

### Conflict of interest

YB and NL are named as inventors on a patent application covering some of the technologies described in this study.

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
