## [Review Process File · Molecular Systems Biology]

Model-guided combinatorial optimization of complex synthetic gene networks

Joerg Schreiber, Ms. Meret Arter, Mr. Nicolas Lapique, Benjamin Haefliger and Yaakov Benenson

Corresponding author: Yaakov Benenson, ETH Zurich

Review timeline:	Submission date:	16 September 2016
	Editorial Decision:	19 October 2016
	Revision received:	18 November 2016
	Accepted:	25 November 2016

Editor: Maria Polychronidou

Transaction Report:

1st Editorial Decision

19 October 2016

Thank you again for submitting your work to Molecular Systems Biology. We have now heard back from the two referees who agreed to evaluate your manuscript. As you will see below, the reviewers raise a number of concerns, which should be carefully addressed in a revision of the manuscript.

The reviewers' recommendations are rather clear so I think that there is no need to repeat the points listed below. In line with comment #3 of reviewer #2 we would ask you to make sure that the miRNA biosensor, its design, intended function etc. as well as the related modeling analyses are described in sufficient detail. The study should be readable/understandable as a standalone paper, without the need to refer to the Mohammadi et al study.

REFeree REPORTS

Reviewer #1:

This paper shows that it is possible to use an approach that combines predictive modeling with diverse construction and screening to obtain synthetic gene networks that work in the way you like. This is an important goal for synthetic biology and the present paper does a solid job in addressing this goal. However the authors need to address the following issues before the work is suitable for publication:

1. There have been previous papers that combine predictive modeling with diverse construction and screening to obtain synthetic gene networks that work in the way you like; notably a 2009 Nature Biotechnology paper by Tom Ellis (Imperial College). The authors need to better acknowledge these earlier efforts and explain how their work is novel compared to these studies. Moreover the authors

should better acknowledge other, earlier computational-experimental studies in synthetic biology, ones that go back to the early days of the field.

2. It would be helpful in the abstract and other places in the paper to better highlight how the good and bad matches between model and experiments were used to derive meaningful mechanistic insights.
3. The Introduction contains many general statements and platitudes about modeling. It would be good if the authors could provide specific examples illustrating specific points and principles.
4. The paper quickly goes from generics and general statements to specifics, dealing with the authors' sensors. A better setup of the specific work that is to be presented would help the reader appreciate what is to come.
5. The authors should discuss how their approach can be extended to other synthetic gene networks. What challenges remain? What are the open questions still out there?
6. The authors should include a figure with a schematic workflow of their approach. It is quite difficult to appreciate how the modeling was coupled with the experiments from the present paper.
7. The schematics in Figure 1 should be expanded and enhanced to give a better sense of the functionality of the synthetic gene networks.
8. In the figures, the authors should more clearly indicate which results are from the modeling and which results are from the experiments.

Reviewer #2:

The authors apply mechanistic modeling to design a genetically encoded biosensor for specific microRNAs, and experimentally construct variants of the designed biosensor to validate the designs' performances. They vary the number and positions of miRNA binding sites, protein-coding exons, and transcriptionally regulated promoters. From their characterization, they find that the developed model was able to explain some aspects of how the miRNA biosensor design affected its function.

Specific Comments

1. In the introduction, the authors mention that metabolic pathway engineering attempts to maximize only a single parameter (yield), but that is not true. For example, pathways are engineered to maximize the product titer (e.g. grams/liter) and organism productivity (grams/liter/hour) as well. These are distinct objectives as high yield pathways may not have high productivity (and vice versa).
2. In the introduction, the authors mention that the prediction of parameter values from DNA sequence is still poorly understood. This is an overly broad statement; it's generally true when engineering genetic systems inside mammalian cells, but much less true when bacterial hosts are used. There are several models that enable one to predict the sequence-expression relationship inside bacteria. For example, the RBS Calculator model is able to predict a bacterial mRNA's translation rate. The authors can further distinguish their approach by highlighting that there are much fewer examples of rational, model-based design when engineering genetic systems in mammalian cells.
3. On pages 3-4, there needs to be a much more complete description of the miRNA biosensor that what is currently presented. The authors should not assume that the reader will have read Mohammadi et. al. prior to reading this manuscript, and therefore the authors need to describe and explain the composition of the miRNA biosensor circuit and its intended function. Currently, a typical reader will have no real understanding of how the biosensor works and the importance of the simulations & design.
4. The authors present a highly detailed comparison between their simulation results and measurements, particularly on the choices of where to position miRNA binding sites. The resulting analysis is very, very specific to the miRNA biosensor characterized here, and it remains uncertain if

there are general design principles that could be "take home" messages to the reader. The authors should highlight the general design principles that could be re-used when engineering a different miRNA biosensor or another entirely different genetic circuit in mammalian cells. The Discussion section does not convey any of this information as the authors have tried to off-load this essential component into a separate publication. Unfortunately, this significantly detracts from the quality of this manuscript.

1st Revision - authors' response

18 November 2016

We would like to thank the Reviewers for their positive assessment of our work, their insightful comments and useful advice. In the point-by-point response below we detail specific changes that we hope will address the concerns and clarify the impact of this study on model-guided circuit engineering and optimization.

Point-by-point response:

Reviewer #1:

This paper shows that it is possible to use an approach that combines predictive modeling with diverse construction and screening to obtain synthetic gene networks that work in the way you like. This is an important goal for synthetic biology and the present paper does a solid job in addressing this goal. However the authors need to address the following issues before the work is suitable for publication:

We thank the Reviewer for their positive assessment of our work and the constructive comments.

1. There have been previous papers that combine predictive modeling with diverse construction and screening to obtain synthetic gene networks that work in the way you like; notably a 2009 Nature Biotechnology paper by Tom Ellis (Imperial College). The authors need to better acknowledge these earlier efforts and explain how their work is novel compared to these studies. Moreover the authors should better acknowledge other, earlier computational-experimental studies in synthetic biology, ones that go back to the early days of the field.

Response 1.1 We thank the Reviewer for the comment and apologize for any omissions in covering the prior art. We have extensively rewritten the introduction to place our work in the context of prior work on modeling and optimization of synthetic circuits. We also acknowledge early works in synthetic biology that used modeling to guide their design efforts.

2. It would be helpful in the abstract and other places in the paper to better highlight how the good and bad matches between model and experiments were used to derive meaningful mechanistic insights.

Response 1.2. In this study, we found no substantial discrepancies between the model and the experiments, and thus our initial mechanistic understanding proved to be correct. Thus, good matches between the predictions and the data, e.g., the fact that improved miRNA knockdown increases the dynamic range, and many other matches referring to the expected trends in On and Off sensor values, are discussed in the relevant places in the manuscript. However we do emphasize in a few places in the manuscript that the data from high-throughput screens and detailed characterization can and should be used to modify the initial model in case of discrepancy.

3. The Introduction contains many general statements and platitudes about modeling. It would be good if the authors could provide specific examples illustrating specific points and principles.

Response 1.3 See also Response 1.1, we have extensively reworked the introduction to make it more concrete, with specific examples showcasing prior work in the field and highlighting the differences between published approaches and this study. We also introduced a Rationale section to explain our reasoning in developing the workflow discussed in the manuscript

4. The paper quickly goes from generics and general statements to specifics, dealing with the authors' sensors. A better setup of the specific work that is to be presented would help the reader appreciate what is to come.

Response 1.4 We have attempted to smoothen this transition by better introducing the rationale behind our workflow (see above), as well as explaining how the sensors work and what their potential usage could be.

5. The authors should discuss how their approach can be extended to other synthetic gene networks. What challenges remain? What are the open questions still out there?

Response 1.5 Both the new “rationale” section and the discussion, as well as the new schematics (new Figure EV1, see point 6 below) are phrased to highlight the general applicability of our optimization strategy.

6. The authors should include a figure with a schematic workflow of their approach. It is quite difficult to appreciate how the modeling was coupled with the experiments from the present paper.

Response 1.6 Thank you for the suggestion, we have created such a scheme as a new Figure EV1.

7. The schematics in Figure 1 should be expanded and enhanced to give a better sense of the functionality of the synthetic gene networks.

Response 1.7 The schematic were extended to clearly show the two logical sensor states (on and off) in terms of component activity levels etc.

8. In the figures, the authors should more clearly indicate which results are from the modeling and which results are from the experiments.

Response 1.8 We have clarified this in the individual legends.

Reviewer #2:

The authors apply mechanistic modeling to design a genetically encoded biosensor for specific microRNAs, and experimentally construct variants of the designed biosensor to validate the designs' performances. They vary the number and positions of miRNA binding sites, protein-coding exons, and transcriptionally regulated promoters. From their characterization, they find that the developed model was able to explain some aspects of how the miRNA biosensor design affected its function.

We thank the Reviewer for their positive assessment of our work and for the constructive comments.

Specific Comments

Comment 2.1. In the introduction, the authors mention that metabolic pathway engineering attempts to maximize only a single parameter (yield), but that is not true. For example, pathways are engineered to maximize the product titer (e.g. grams/liter) and organism productivity (grams/liter/hour) as well. These are distinct objectives as high yield pathways may not have high productivity (and vice versa).

Response 2.1: we thank the Reviewer for this comment; indeed this has been an omission on our side. The statement has been rectified and a reference was added pointing to those different optimization factors.

2. In the introduction, the authors mention that the prediction of parameter values from DNA sequence is still poorly understood. This is an overly broad statement; it's generally true when engineering genetic systems inside mammalian cells, but much less true when bacterial hosts are used. There are several models that enable one to predict the sequence-expression relationship inside bacteria. For example, the RBS Calculator model is able to predict a bacterial mRNA's translation rate. The authors can further distinguish their approach by highlighting that there are much fewer examples of rational, model-based design when engineering genetic systems in mammalian cells.

Response 2.2 We thank the Reviewer for the suggestion; we have modified the introduction to better reflect the state of the art in predicting function from sequence. We distinguish between de novo predictions and those based on large dataset combined with machine learning.

3. On pages 3-4, there needs to be a much more complete description of the miRNA biosensor that what is currently presented. The authors should not assume that the reader will have read Mohammadi et. al. prior to reading this manuscript, and therefore the authors need to describe and explain the composition of the miRNA biosensor circuit and its intended function. Currently, a typical reader will have no real understanding of how the biosensor works and the importance of the simulations & design.

Response 2.3 We have added an explanation about the potential utilization of miRNA classifier networks, and added a description of sensor mechanism of operation. Figure 1A was also expanded to illustrate the configuration of different sensor states.

4. The authors present a highly detailed comparison between their simulation results and measurements, particularly on the choices of where to position miRNA binding sites. The resulting analysis is very, very specific to the miRNA biosensor characterized here, and it remains uncertain if there are general design principles that could be "take home" messages to the reader. The authors should highlight the general design principles that could be re-used when engineering a different miRNA biosensor or another entirely different genetic circuit in mammalian cells. The Discussion section does not convey any of this information as the authors have tried to off-load this essential component into a separate publication. Unfortunately, this significantly detracts from the quality of this manuscript.

Response 2.4 We thank the Reviewer for raising this point. This "take home message" is an important component of the study. First, we note that every circuit family has its own unique features and therefore a model of a given circuit, and the conclusions thereof, might be only applicable to the specific family under investigation or closely-related families. In our case, the circuit family is a particular double inversion topology (See Fig. 1A top panel, on the right). Regardless of what the inputs and the intermediate components are, the repression steps must be very efficient and triggered by low amounts of either the input or the repressor; at the same

time the induction should operate far from saturation. This is the immediate take home message of this study. The broad importance of the work is to demonstrate the workflow that can be applicable to diverse circuit families: introducing a performance metric and evaluating multiple parameter combinations on a grid, and uncovering trends, correlations and cross-correlations to reveal optimal parameter regimes. Next, in the experimental part, designing a library of diverse components that span the parameter phase space and combinatorial screening of multiple circuit variants in an effort to reach the favorable parameter regime and either validate or adjust the original model. The manuscript has been modified to convey these ideas in the Introduction, the Rationale and the Discussion sections, and we hope the general applicability of our study comes across much better now.

2nd Editorial Decision

25 November 2016

Thank you for sending us your revised manuscript. We are satisfied with the modifications made and I am pleased to inform you that your paper has been accepted for publication in Molecular Systems Biology.

Corresponding Author Name: Yaakov Benenson

Journal Submitted to: MSB

Manuscript Number: MSB-16-7265R